# Influence of Alkaline Treatment on Structural Modifications of Chlorophyll Pigments in NaOH—Treated Table Olives Preserved without Fermentation

**DOI:** 10.3390/foods9060701

**Published:** 2020-06-01

**Authors:** Marta Berlanga-Del Pozo, Lourdes Gallardo-Guerrero, Beatriz Gandul-Rojas

**Affiliations:** Chemistry and Biochemistry of Pigments, Food Phytochemistry, Instituto de la Grasa (CSIC), Campus Universitario Pablo de Olavide, Edificio 46, Ctra. Utrera km 1, 41013 Sevilla, Spain; mberlanga@ig.csic.es (M.B.-D.P.); lgallardo@ig.csic.es (L.G.-G.)

**Keywords:** chlorophyll, pigments, allomerization, table olive, alkaline treatment, phytyl-chlorin, phytyl-rhodin

## Abstract

Alkaline treatment is a key stage in the production of green table olives and its main aim is rapid debittering of the fruit. Its action is complex, with structural changes in both the skin and the pulp, and loss of bioactive components in addition to the bitter glycoside oleuropein. One of the components seriously affected are chlorophylls, which are located mainly in the skin of the fresh fruit. Chlorophyll pigments are responsible for the highly-valued green color typical of table olive specialties not preserved by fermentation. Subsequently, the effect on chlorophylls of nine processes, differentiated by NaOH concentration and/or treatment time, after one year of fruit preservation under refrigeration conditions, was investigated. A direct relationship was found between the intensity of the alkali treatment and the degree of chlorophyll degradation, with losses of more than 60% being recorded when NaOH concentration of 4% or greater were used. Oxidation with opening of the isocyclic ring was the main structural change, followed by pheophytinization and degradation to colorless products. To a lesser extent, decarbomethoxylation and dephytylation reactions were detected. An increase in NaOH from 2% to 5% reduced the treatment time from 7 to 4 h, but fostered greater formation of allomerized derivatives, and caused a significant decrease in the chlorophyll content of the olives. However, NaOH concentrations between 6% and 10% did not lead to further time reductions, which remained at 3 h, nor to a significant increase in oxidized compounds, though the proportion of isochlorin *e*_4_-type derivatives was modified. Chlorophyll compounds of series *b* were more prone to oxidation and degradation reactions to colorless products than those of series *a*. However, the latter showed a higher degree of pheophytinization, and, exclusively, decarbomethoxylation and dephytylation reactions.

## 1. Introduction

Table olives may be considered one of the most nutritious and least caloric snacks, thanks to their balanced fat composition, in which monounsaturated oleic acid predominates and includes essential fatty acids, and to their fiber, vitamin and mineral content [1,2,3]. In addition, table olives contain phytochemicals such as polyphenols [4], chlorophylls, carotenoids [5], and triterpenic acids [6], which give them functional value. In fact, the table olive is recognized as an essential component of the Mediterranean diet, having been explicitly included in the second level of its nutritional pyramid [7] as an aperitif or culinary ingredient, with a recommended daily consumption of one or two portions (15–30 g). According to the International Olive Council [8], worldwide table olive consumption for the 2018/2019 season will be around 2,667,000 t, which means an increase of around 21% in the last 10 years.

Unlike most fruits, olives must be processed to be edible, since the fresh fruit has an extremely bitter taste, due to its high phenolic compound content, mainly oleuropein. These bitter components can be totally or partially eliminated by various procedures including both hydrolysis—chemical and/or enzymatic—and brine diffusion mechanisms [9]. Of all the procedures, alkaline treatment is the most widely used [10], since it is the method applied to olives processed in the Spanish or Sevillian style (green table olives) and in the Californian style (black table olives), which are two of the main commercial table olive preparations worldwide. It is also the de-bittering method used in the preparations called green ripe olives, which are widely consumed in the United States, as well as in other table olive processing specialties—the Castelvetrano, Picholine and Campo Real styles [11].

Alkaline treatment is a key stage in the preparation of table olives, and can be more or less intense depending on factors such as the variety and state of maturity of the olive, the temperature and quality of the water [9], as well as the subsequent preservation system. In the processing of Spanish-style green table olives, the fruits are treated with a diluted solution of NaOH in water, with a concentration between 2% and 5%, until this solution (lye) penetrates two thirds or three quarters of the pulp towards the stone, which usually takes 4–11 h [12]. Subsequently, after several washings with water, the olives are put in brine (NaCl solution), where they undergo an acid-lactic fermentation. In the preparations of olives in the Castelvetrano [13,14], Picholine [9], Campo Real [15] and green ripe styles [16,17], the fruits remain in alkaline conditions for longer, and the lye can penetrate to the stone. After alkaline treatment and washing, the fruits are kept in refrigerated conditions and/or subjected to heat treatment, thus avoiding any fermentation process.

During the preparation of the olives, the fruits undergo changes in their composition that are mediated by the different stages involved in the processing system. Thus, the way in which the alkaline treatment is applied, the number and time of subsequent washings of the fruits, the presence or otherwise of a fermentation stage, etc., will have an impact on the end product. The treatment with NaOH causes structural modifications in both the epicarp and the mesocarp of the olive, which will depend on the concentration and temperature of the alkaline solution, and which influence the physical-chemical composition of the fruit [18]. The lye solution that penetrates the pulp hydrolyses the oleuropein and ligstroside, producing non-bitter hydrolyzed phenols such as hydroxytyrosol and tyrosol. In addition, this alkaline solution changes the composition of the polysaccharides in the cell wall structure, reducing the firmness of the fruit [19]. The higher the concentration of the lye, and the longer the treatment, the greater the loss of firmness. Chemical damage to the olive’s skin, and to its cell structure, allows for more rapid diffusion of the remaining phenolic compounds, and of the sugars, into the brine during the subsequent rinsing and fermentation stages.

Different treatment options with alkaline solutions have been studied to improve the organoleptic quality of the end product and the characteristics of the washing water. Transporting mechanically-harvested olives in low-grade lye, rather than resting prior to processing, avoids peeling and superficial dark spots (damage). On the other hand, the addition of calcium and/or sodium salts to the alkaline lye, and cooling it to 8 °C, gives rise to treatments that improve the texture and prevent the breakage of the olive skin [12,20], while the replacement of NaOH by KOH improves the potential use of the washing waters for agronomic purposes [21]. An inappropriate alkaline treatment—low NaOH concentration and/or insufficient alkaline penetration—leads to the presence of antimicrobial compounds in the brine, which inhibits the growth of *Lactobacillus pentosus*, negatively affecting the fermentation processes [22].

Alkaline treatment and subsequent washing lead to a high loss of volatile compounds [23] and bioactive substances in the olives, such as phenolic compounds and triterpenic acids, which are diffused from the fruit into the wastewater [6,24]. In this regard, García et al. [25] recently verified that black ripe olives produced in the USA have lower contents in phenolic compounds and triterpenic acids than those produced in Spain, and they attribute these differences to the alkaline treatment used in the former, which involves a higher number of alkali/washing cycles.

Chlorophylls (*a* and *b*) and carotenoids are the pigments responsible for the color of the olives in their green ripening state, and constitute another group of phytochemicals in olives that are affected by their processing: they undergo certain structural transformations that can have an impact on both the color and the functional value of the end product. All green table olive preparations are made with fruits of the same chlorophyll and carotenoid composition; however, the transformation these compounds undergo is different for each processing system, depending on whether or not they are treated with alkali and/or fermented. Thus, in each case, an end product with its own composition of pigments and a characteristic color will be obtained [26,27].

The transformations of chlorophylls and carotenoids during the processing of Spanish-style green table olives, which includes alkaline treatment and fermentation, have been widely studied [28,29,30,31]. However, for specialties of table olives treated with alkali and preserved without fermentation, only olives processed in the Castelvetrano style have been studied [27,32]. One of the main and most highly-valued characteristics of these specialties of table olives is a typical bright green color. For some years now, suspicions of the color adulteration of table olives by regreening practices—by addition of E141ii coloring additive or Cu^2+^ salts—have arisen. In this sense, several studies have been carried out aimed to characterize the chlorophyll pigment profile of commercial bright green table olives [33,34,35,36].

In general, it is known that alkaline treatment causes the partial degradation of chlorophylls *a* and *b* into more hydrosoluble derivatives, but with a green color similar to that of their respective precursors. Depending on the conditions in which this treatment is carried out—essentially the volume of fruits treated and the presence of oxygen—dephytylation reactions of the chlorophylls by the activation of endogenous chlorophyllase may be caused, and/or oxidation reactions affecting the isocyclic ring of the chlorophyll structure (allomerization reactions), originating mainly phytyl-chlorin or phytyl-rhodin derivatives, depending on whether they are from series *a* or *b*, respectively [5,27]. The carotenoid pigment fraction, however, is not affected, since they are alkali-stable compounds [37]. There are many ways in which alkaline treatment can be applied to olives, even reusing a lye solution to treat several batches. This is a very widespread practice aimed at reducing the polluting environmental impact of NaOH solutions [10,38]. In this regard, Gallardo-Guerrero et al. [39] investigated, for a fixed treatment time, the influence of three factors: concentration of NaOH, use of recycled alkaline solution and fruit size. It was shown that the use of recycled solution significantly intensified the oxidizing capacity of such treatment, to a greater extent than the concentration of alkali, and fruit size had a certain effect, with the greatest transformation in the smaller ones.

The aim of this study was to advance knowledge of the complex action that the alkaline treatment of olives has on the structural modifications of chlorophylls, bioactive components [40,41] and pigments responsible for the characteristic bright green color of table olives not preserved by fermentation. Specifically, the effect of nine combinations of two important parameters of the alkali treatment (NaOH concentration and treatment time) on green table olives processed in the Campo Real style—with penetration of the alkaline solution as far as the stone—and preserved for one year under refrigeration conditions, was investigated. A direct relationship between the intensity of the alkali treatment and the degree of degradation of the chlorophylls was shown.

## 2. Materials and Methods 

All procedures were performed under dimmed green light to avoid any photo-oxidation of chlorophylls.

### 2.1. Raw Material and Preparation of Samples

The study was carried out on olives of the Verdial variety (*Olea europaea* L.) collected from olive trees in an orchard located in Paterna del Campo (Huelva, Spain). Around 5 kg of fruits were picked at the end of August in the intense green ripening stage. Green table olives were processed at laboratory scale, according to the Campo Real-style [15]. The experimental design consisted in the treatment of the fruits with an alkaline solution that included 3% NaCl, and NaOH in a concentration equal to or greater than 2%. Glass containers (370 mL of capacity) were filled approximately with 200 g of fruits and 150 mL of alkaline brine. Nine samples were prepared, increasing in the NaCl solution the percentage of NaOH from 2% to 10% (Table 1). Treatment time was adjusted for each sample in order to achieve a penetration of the alkaline solution into the fruits until reaching the stone. Longitudinal cuts were carried out on the fruits every 30 min for controlling the alkaline penetration. It was visible to the naked eye by the turn of the flesh greenish color to dark brown. The applied times varied from 3 h, in those samples processed with alkaline solutions of NaOH concentration ≥6% (Table 1, samples E, F, G, H and I), to 7 h in the sample treated with the lowest concentration of NaOH (sample A), which was considered the standard treatment.

After alkaline treatment, olive fruits were washed twice with tap water. The first one was dynamic under running water, and the second by immersion for 20 h. Finally, the fruits were placed in 6% NaCl brine and kept for 12 months in a refrigerated chamber at 4 °C to avoid fermentation.

### 2.2. Chemicals and Standards

Ammonium acetate was supplied by Fluka (Zwijndrecht, The Netherlands). Solvents used for chromatography were HPLC grade (Prolabo, VWR International Eurolab, Barcelona, Spain). Analysis grade solvents were supplied by Scharlau (Microdur, Sevilla, Spain). The deionized water was obtained from a Milli-Q^®^ 50 system (Millipore Corporation, Milford, MA, USA). For all purposes, analytical grade (American Chemical Society) reagents were used (Merck, Madrid, Spain). Standards of chlorophylls *a* and *b* were supplied by Sigma Chemical Co. (St. Louis, MO, USA), and standards of pheophytin *a* and pyropheophytin *a* were provided by Wako chemicals Gmbh (Neuss, Germany). Standards of pheophorbide *a*, pyropheophorbide *a*, and chlorine *e*_6_ and rhodin *g*_7_ sodium salts, were purchased from Frontier Scientific Europe Ltd. (Carnforth, Lancashire, UK). The C-13 epimers (chlorophylls *a*’ and *b*’) were prepared by treatment of the respective chlorophyll with chloroform, and 13^2^-OH-chlorophylls (*a* or *b*) were obtained by selenium dioxide oxidation of the corresponding chlorophyll at reflux-heating for 4 h in pyridine solution under argon [42]. Pheophytin *b* was prepared from a solution of chlorophyll *b* in ethyl ether by acidification with 13% HCl (v/v), and shaking the mixture for 5 min [43]. Standards purity, evaluated by HPLC, was ≥95% in all cases with the exception of rhodin *g*_7_ sodium salt (~90%).

### 2.3. Pigment Extraction

Previously to the pigment analysis, the fruits were washed several times by immersion in water until the pH of the wash water was neutral. Pigments were extracted with N,N-dimethylformamide (DMF) according to the method of Mínguez-Mosquera and Garrido-Fernández [44], slightly modified as described in detail in Gandul-Rojas et al. [34]. The technique is based on the selective separation of pigments and lipids between DMF and hexane, respectively, which allows obtaining a fat–free pigment extract. From a homogenized triturate prepared with 5 destoned fruits (ca. 30 g), two samples of 2 g each were weighed to carry out the pigment extraction in duplicate. In this methodology, the fat-free pigments dissolved in the DMF phase are subsequently transferred to hexane/diethyl ether (1:1, v/v) mixture by adding a cold solution (4 °C) of 10% NaCl. In this way, the pigment extract may be concentrated to dryness without exceeding 30 °C. The dry residue was dissolved in 1 mL of acetone for its subsequent analysis by HPLC. Similarly, solvent of the hexane phase was evaporated and the remaining residue eluted in a known volume of hexane. In this way, casual losses of pheophytin *a* in the hexane phase can be quantified by direct absorbance measurement at 670 nm using the molar absorption coefficient *EmM* = 53.4.

### 2.4. Pigment Analysis by HPLC

Separation, identification and quantification of pigments were carried out by HPLC (HP 1100 Hewlett-Packard, Palo Alto, CA; fitted with an HP 1100 automatic injector and diode array detector). A stainless-steel column (20 × 0.46 cm i.d.), packed with a multifunctional endcapped deactivated octadecylsilyl (C18) Mediterranea™ Sea18, 3 μm particle size (Teknokroma, Barcelona, Spain) was used. The column was protected by precolumn (1 × 0.4 cm i.d.) packed with the same material. Solutions of pigment extract were centrifuged at 13,000× *g* prior to injection into the chromatograph. Pigment separation was performed using an elution gradient (flow rate 1.250 mL·min^−1^) with the mobile phases (A) 0.5 M ammonium acetate in water/methanol (1/4, v/v) and (B) methanol/acetone (1/1, v/v). The gradient scheme is a modification of that of Mínguez-Mosquera et al. [45], as previously described by Gandul-Rojas and Gallardo-Guerrero [32]. The on-line UV-Vis spectra were recorded from 350 to 800 nm with the photodiode-array detector. Data were collected and processed with a LC HP ChemStation (Rev.A.05.04). Pigments were identified by co-chromatography with the corresponding standard and from the spectral characteristics as described in detail in previous publications [27,33,46].

Spectrophotometric detection of pigments was performed by absorbance at different wavelengths. For each pigment, the wavelength closest to its absorption maximum in the red region was chosen: 626 nm for the Mg complex of 15^2^-Me-phytyl-rhodin *g*_7_ ester; 640 nm for the Mg complex of both 15^2^-Me-phytyl-chlorin *e*_6_ and 15^2^-Me-phytyl-isochlorin *e*_4_ esters; 650 nm for chlorophylls *b* and *b*’, 13^2^-OH-chlorophyll *b* and the Mg-free chlorophyll derivatives of the series *b*; and 666 nm for chlorophylls *a* and *a*’, 13^2^-OH-chlorophyll *a* and the Mg-free chlorophyll derivatives of the series *a*.

Pigments were quantified using external standard calibration curves (amount versus integrated peak area) prepared with the pigment standards listed in Section 2.2. Calibration curves for chlorophylls *a* and *b* were used for their respective epimers and 13^2^-OH-derivatives. Calibration curve obtained for pheophytin *a* was used for pheophytin *a*’. For pigments with chlorin- and rhodin-type structures, calibration curves obtained for chlorine *e*_6_ and rhodin *g*_7_ sodium salts were used, respectively. The calibration equations were obtained by least-squares linear regression analysis over a concentration range according to the levels of these pigments in green table olives. Injections in duplicate were made for five volumes of each standard solution (range of concentrations between 2 and 2500 ng; R^2^ < 0.9983). Limit of detection (LOD) and limit of quantification (LOQ), defined as a signal-to-noise ratio of 3.3 and 10, respectively, were LOD 0.30–1.19 ng and LOQ 0.90–3.6 ng.

### 2.5. Statistical Analysis

Analyses in this study were performed in duplicated. Statistica software for Windows (version 6, StatSoft, Inc., Tulsa, OK, USA, 2001) was used for data processing. Data were expressed as mean values ± standard deviation (SD). The data were analyzed for differences between means using one-way analysis of variance (ANOVA). Post hoc comparisons were carried out according to Duncan’s multiple range-test and the differences were considered significant when *p* < 0.05.

## 3. Results and Discussion

The alkaline treatments, which the olives were subjected to, caused—to a greater or lesser extent—the transformation of the chlorophylls *a* and *b* present in the fresh fruit. Figure 1 shows the HPLC chromatograms resulting from the separation of pigments from the olives, before (fresh fruit) and after having been subjected to alkaline treatment. All the alkali-treated samples showed a similar qualitative pigment profile, and the chromatogram corresponding to sample A (standard alkaline treatment) was selected by way of representation. Table 2 shows the chromatographic and spectroscopic characteristics of the different pigments identified, and Figure 2 shows the structure of each.

Once olive fruits were alkali-treated, a greater presence of chlorophyll epimers (*a*’ and *b*’) regarding their respective precursor was evidenced. Moreover, as expected from previous studies [27,32], chlorophylls *a* and *b* underwent allomerization reactions after the alkaline treatment of the olives, which produced, as well as the hydroxylated derivatives—13^2^-OH-chlorophyll *a* and 13^2^-OH-chlorophyll *b*—oxidized chlorophyll derivatives of the chlorin- and rhodin-types, which are characterized by their open isocyclic ring (V) of the chlorophyll structure (IR 2 in Figure 2). In series *a*, the compounds Mg-15^2^-Me-phytyl-chlorin *e*_6_ ester and Mg-15^2^-Me-phytyl-isochlorin *e*_4_ ester were detected, whose structures differ only in the C-13 substitute, which in the former is a carboxyl group, while in the latter it is a hydrogen. With respect to series *b*, Mg-15^2^-Me-phytyl-rhodin *g*_7_ ester was detected. In addition to these allomerization reactions, evidence of pheophytinization reaction was found, by substitution of the Mg ion by 2 H in the porphyrin ring, which gave rise to the formation of the magnesium–free derivatives of the previous compounds, i.e., the esters 15^2^-Me-phytyl-chlorin *e*_6_ and 15^2^-Me-phytyl-isochlorin *e*_4_ in series *a*, and 15^2^-Me-phytyl-rhodin *g*_7_ in series *b*. This modification also affected the original chlorophylls *a* and *b*, with the formation of pheophytins *a* and *b*, respectively. Although the pheophytinization reaction typically occurs under acidic conditions, it is also fostered by temperature increase [47]. Therefore, the heat generated during the alkaline treatment of the fruits, due to the alkaline hydrolysis reactions produced by the hydroxyl ions diffusing inside the olive [48], must have fostered the pheophytinization reaction. This same factor even caused decarbomethoxylation due to loss of the -COOCH_3_ group at C-13^2^, since pyroderivatives (pyropheophytin *a* and pyropheophorbide *a*) were also produced. The dephytylated derivatives pheophorbide *a* and the previously mentioned pyropheophorbide *a* were also detected, though at much lower proportion. In plant foods, pheophytins, pheophorbides and pyroderivatives are the chlorophyll compounds that originate widely during thermal processing or fermentation of fruits, vegetables and green algae [47] while chlorophyll derivatives with phytyl-chlorin or phytyl-rhodin structures are specifically associated with the processing of alkali-treated table olives [26,27,31]. In another field, this type of derivatives has also been identified during the senescence of microalgaes incubated under oxic conditions [49], and more recently as secondary metabolites of digestion in mice from a diet rich in chlorophylls [50]. Its presence in the liver indicates the existence of alternative metabolic pathways that modify the structure of the chlorophyll macrocycle increasing its polarity and, as indicated by the authors [50], to facilitate probably a greater metabolism and/or excretion.

As mentioned above, the profile of chlorophyll derivatives identified in the olives was practically the same in all the treatments tested. However, differences were found both in the total chlorophyll pigment content of the fruits and in the proportion of each of the derivatives after the alkaline treatment, with some of the precursor pigments even disappearing in the more intense procedures (Table 3). The outstanding presence of oxidized chlorophyll derivatives showed that the alkaline treatment was responsible for their formation in the fruit processing, as has been shown in previous studies carried out in mildly alkaline conditions [46]. Sample A, which had been subjected to the mildest alkaline treatment conditions, and normally used in the preparation of olives in the Campo Real style, did not show significant variation with respect to the total amount of chlorophyll pigments present in the fresh fruit (*p* < 0.05). However, for the rest of the samples, as the olives were subjected to a more intense alkaline treatment, in general, lower pigment content was found, which showed a transformation of the chlorophylls into colorless products (Figure 3). Thus, the increase in NaOH concentration from 2% to 3% caused about 16% destruction of chlorophyll pigments in the fruits (sample B), despite the fact that the treatment time required was reduced from 7 to 6 h. This effect was much more pronounced in sample C, which had been treated with 4% NaOH. The process was shortened to 5 h but a significant decrease of 68% in pigment content was noted, compared to sample B. However, the effects produced on the total chlorophyll content of the olives by the treatments with 5% and 6% NaOH, with time reduction at 4 and 3 h, respectively (samples D and E), were not significantly different from those of sample C. The alkaline treatments from sample F, in which the NaOH concentration was increased from 7% to 10% (samples F–I), did not lead to further reductions in the time needed for the total penetration of the alkaline lye, which was maintained at 3 h. In these samples, the quantity of pigments continued to decrease more gradually and, in some cases, not significantly.

In a previous study, Gandul-Rojas and Gallardo-Guerrero [32] noticed that the chemical treatment of olives with NaOH transformed the different chlorophyll pigments into their allomerized derivatives characterized by an open isocyclic ring, and that the degree of this transformation increased when the contact time between the fruit and the alkaline brine was prolonged. In the present study, olive fruits were treated with different concentrations of NaOH. In each sample, the fruits were maintained in the alkaline solution for the time needed to achieve the same alkali penetration in all of them—this was until the fruit stone was reached, which is characteristic for the processing of Campo Real-style table olives [15]. Under these conditions, the treatment time could be excluded as an independent variable, allowing the in-detail analysis of the influence of NaOH concentration on modifications of chlorophyll compounds. The results showed that the standard treatment (sample A), caused the formation of chlorophyll derivatives with open isocyclic ring by 12%, while the rest of the pigments (88%) maintained a closed isocyclic ring structure (Figure 4).

The increase of 1% in the concentration of NaOH in the alkaline brine, in spite of reducing the necessary treatment time by 1 h, caused an increase of 37% in the amount of these oxidized derivatives present in the olives (sample B). From the treatment of the fruits with alkaline brine of NaOH concentration equal or higher than 4% (samples C–I), the percentage of open isocyclic ring chlorophyll derivatives was already higher than the corresponding percentage of closed isocyclic ring derivatives, reaching around 73% in most of the samples, and without significant differences (*p* < 0.05) among them. This result suggested that there was a concentration of NaOH above which an increase in the strength of the alkaline treatment did not lead to detecting higher percentages of oxidized chlorophyll derivatives with open isocyclic ring (Figure 4), although degradation to colorless products did occur (Figure 3). Therefore, it was observed that the samples subjected to the mildest treatments (samples A–C), were those which showed the greatest differentiation, both in the total chlorophyll pigment content, and in the relationship between the chlorophyll derivatives with closed and open isocyclic ring, with no significant differences being found in this relationship from sample C.

Among the chlorophyll compounds with open isocyclic ring, two groups of derivatives could be distinguished according to the R_5_ substituent of their structure (Figure 2). One of them was formed by the chlorophyll derivatives with a -COOH group in R_5_, that is, those with chlorine *e*_6_- and rhodin *g*_7_-type structures. The other group included the derivatives in which the R_5_ substitute was an H, corresponding, therefore, to compounds with isochlorin *e*_4_-type structure. To facilitate an understanding of the results, the opening of the isocyclic ring was referred to as type O (O-ring), for the first group and type Iso (Iso-ring), for the second. Figure 5 shows the percentage composition represented by each of these groups of derivatives, with respect to the total content of chlorophyll compounds.

The number of Iso-ring derivatives was minimal (less than 1%), in samples A and B, while that of O-ring derivatives was 11% and 36%, respectively. However, from the C sample—NaOH treatments with alkaline brine of concentration ≥4%—the presence of Iso-ring compounds increased considerably, being in some cases equal to or higher than those of O-ring (samples C–E), and reaching about 42% of the total chlorophyll compounds. However, from the E sample, a certain tendency to decrease the proportion of the Iso-ring derivatives was observed, as the NaOH concentration increased, but no parallel decrease of the O-ring derivatives was noticed. It is likely that the latter are precursors of the Iso-ring derivatives, and these, in turn, represent the step prior to the formation of colorless products, detected in the samples treated with higher NaOH concentration (Figure 3).

On the other hand, the influence of the alkaline treatment on the degradation of the chlorophyll pigments was evaluated, depending on whether they were of series *a* or *b*. To this end, the data were calculated with respect to the total content of each of the series (Figure 6). It was observed that the evolution of the formation of chlorophyll derivatives with open isocyclic ring was similar for both series, increasing as the concentration of NaOH in the alkaline brine increased, as had been previously seen globally (Figure 4). However, it was noted that, in all the samples, the percentage of oxidized derivatives corresponding to series *b* was greater than that of series *a*, reaching 100% in sample I, as opposed to the 62% quantified for series *a*. However, in none of the samples were detected Iso-ring derivatives of series *b* (Table 3) which, as mentioned above, are likely to be the step prior to the formation of colorless products. At the same time, it was found that the total content of chlorophyll derivatives of series *b* decreased in a much higher proportion than those of series *a* in most of the samples. Therefore, it could be that the Iso-ring derivatives of the series *b* were immediately transformed into uncolored compounds, making their detection impossible. Already in sample C, the series *b* derivatives decreased by about 90%, with respect to the initial fresh fruit content, fluctuating this value between 82% and 95%, in the D–I samples. In series *a*, on the other hand, the total content of chlorophyll derivatives decreased only 63% in sample C, remaining around this value until sample G. It was in samples H and I, where the degradation of derivatives of series *a* reached values close to those of series *b*, although they continued to be lower. As regards the hydroxylated derivatives (13^2^-OH-chlorophyll *a* and 13^2^-OH-chlorophyll *b*), a significantly higher percentage was also observed in series *b* (12%–21%) than in series *a* (1%–4%) (Table 3). All these results pointed to higher sensitivity in series *b* than series *a*, for the transformations and/or degradations of the isocyclic ring, caused by the alkaline treatment. A similar result was previously found during processing of Castelvetrano-style table olives [27].

However, in the case of the substitution reaction of Mg by 2 H in the porphyrin ring (pheophytinization), the result was the opposite, as expected from previous kinetic studies carried out with chlorophylls *a* and *b* [47], and in real fermented olive system [29]. This reaction was much more pronounced in all the chlorophyll compounds of series *a*, both in the chlorophyll (closed isocyclic ring), and in the derivatives with open isocyclic ring. Overall, for series *a*, Mg-free derivatives were recorded at from 37% in sample A to 90% in sample C, and varying in the rest of the samples between 61% and 80%, although without following any particular pattern. In series *b*, however, pheophytinization was not generalized, and was detected only in certain samples (A, B, C and D), and at a much lower proportion than in series *a* (Figure 6), with a maximum value of 32% recorded in sample B.

Likewise, in series *a* it was noted that part of the initial chlorophyll (closed isocyclic ring structure), which had not been transformed by the alkaline treatment into allomerized derivatives, was transformed into Mg-free derivatives. This transformation represented a greater proportion as the NaOH concentration in the alkaline brine increased, to the extent that the precursor pigment, chlorophyll *a*, was not detected in sample I (Figure 6a). On the other hand, in the derivatives with open isocyclic ring, this pattern was somewhat different. A higher percentage of Mg-free compounds was recorded in samples A to E, and the derivatives with and without Mg remained at similar percentages from sample F (Figure 6b). With respect to series *b*, in general, only the Mg-free derivative with open isocyclic ring was detected, except in the sample that had a higher percentage of this compound (sample B), in which a small amount of the Mg-free derivative with closed isocyclic ring (pheophytin *b*) was also detected (Figure 6c,d).

The alkaline treatment of the olives also caused the dephytylation reaction at C-17^3^ in the chlorophyll derivatives of series *a*, with a minority presence of pyropheophorbide *a* in most samples. It highlighted that, in general, this was the only dephytylated derivative found, with the exception of sample B, in which a small amount of pheophorbide *a* was also detected (Table 3). The recorded percentages of pyropheophorbide *a* are shown in Figure 7. In samples A–F they varied between 0% and 1%, without following any particular pattern, and increased slightly to 2% in samples G and H, and more markedly in sample I, in which it reached 11%. Dephytylated chlorophyll derivatives can be formed by an enzymatic route, through the action of the endogenous enzyme chlorophyllase, or by a chemical route, through the non-specific acid or alkaline hydrolysis of esters [46]. However, according to Mínguez-Mosquera and Gandul-Rojas [46], the limited presence of oxygen in the alkaline medium may foster a certain degree of specific de-esterification of phytol by chemical action, limiting other parallel oxidation reactions at C-13. In this study, the observed increase of pyropheophorbide *a* from sample G, treated with an alkaline brine of 8% NaOH, did not seem, in principle, to be related to the activity of chlorophyllase, since, although this enzyme has optimal activity at pH 8.5 [51], strongly alkaline values promote its destabilization [52]. By contrast, since other reaction conditions were equal, the increase in NaOH concentration could foster specific chemical hydrolysis, as discussed before, especially in sample I. Nevertheless, and given the demonstrated thermal stability of chlorophyllase [51], it cannot be ruled out that the heat generated by the hydrolytic reactions inside the fruit could have fostered competition between thermal activation of chlorophyllase, and strongly alkaline pH destabilization, resulting in an initial enzymatic formation of dephytylated derivatives

On the other hand, the presence of pyropheophorbide *a* and pyropheophytin *a* in all the samples showed that, during the treatment of the olives with alkali, in series *a*, there was also the decarbomethoxylation reaction at C-13^2^, which led to the formation of the pyroderivatives. In general, this reaction was more pronounced during the more intense alkaline treatments (Figure 7). In sample A, which was subjected to the standard alkaline treatment, only 2% pyroderivatives was produced, a percentage that increased to 11% in sample B, and up to values between 20% and 25%, without a fixed pattern or significant differences, in samples C–G. In alkaline treatments with more concentrated NaOH solutions, samples H and I, up to 26% and 31% pyroderivatives, respectively, were recorded. The formation of pyroderivatives is normally associated with the heat treatment of vegetables. Tarrado-Castellarnau et al. [48] showed an increase in the internal temperature of fruits during the alkaline treatment of green olives. The characterization of the heat transfer process led them to hypothesize that this increase could only be the result of the heat released inside the fruit, as a result of alkaline hydrolysis reactions and, to a lesser extent, the dilution of the solution with the water in the pulp. This heat generated must be the origin of the pyroderivatives, and a higher reach of the hydrolysis reactions of all the components of the olives, as the concentration of NaOH in the alkaline solution was increased, might explain the greater formation of them in olives subjected to more intense treatments.

## 4. Conclusions

All the treatments caused various types of reactions in the chlorophylls, oxidation with an opening of the isocyclic ring being the main one, as well as pheophytinization and degradation to colorless products. To a lesser extent, decarbomethoxylation and dephytylation reactions were detected. The increase in NaOH concentration from 2% to 5% reduced the time needed for the total penetration of alkaline brine from 7 to 4 h, but fostered greater formation of chlorophyll derivatives with open isocyclic ring, and caused a significant decrease in the chlorophyll content of the olives. However, NaOH concentrations between 6% and 10% did not lead to further reductions in the treatment time, which remained at 3 h, nor to a significant increase in oxidized compounds, although the proportion of derivatives with open isocyclic ring of isochlorin *e*_4_ structure was modified, suggesting that these compounds might represent the stage prior to the formation of colorless products.

The chlorophyll compounds of series *b* were more sensitive than those of series *a* to the isocyclic ring oxidation reactions caused by the alkaline treatment, as well as to the degradation to colorless products. However, series *a* showed a higher degree of pheophytinization and, exclusively, decarbomethoxylation and dephytylation reactions. The first two transformations were fostered by the heat generated inside the fruit as a result of alkaline hydrolysis reactions. Dephytylation in a small proportion of chlorophylls could be the result of alkaline hydrolysis of phytol, under limited oxygen conditions, and/or thermal activation of chlorophyllase, prior to their destabilization due to a highly alkaline pH.

Therefore, a direct relationship between the degradation of chlorophyll pigments and the intensity of alkali treatment in the processing of green table olives was evidenced, with losses of more than 60% being quantified at NaOH concentration of 4% or higher. In spite of the advantage that this increase entails in reducing the necessary treatment time, the parallel negative effect on the intensity of the green color, and the functional value of the product, must be taken into account when optimizing the efficiency of the process.

## Figures and Tables

**Figure 1 foods-09-00701-f001:**
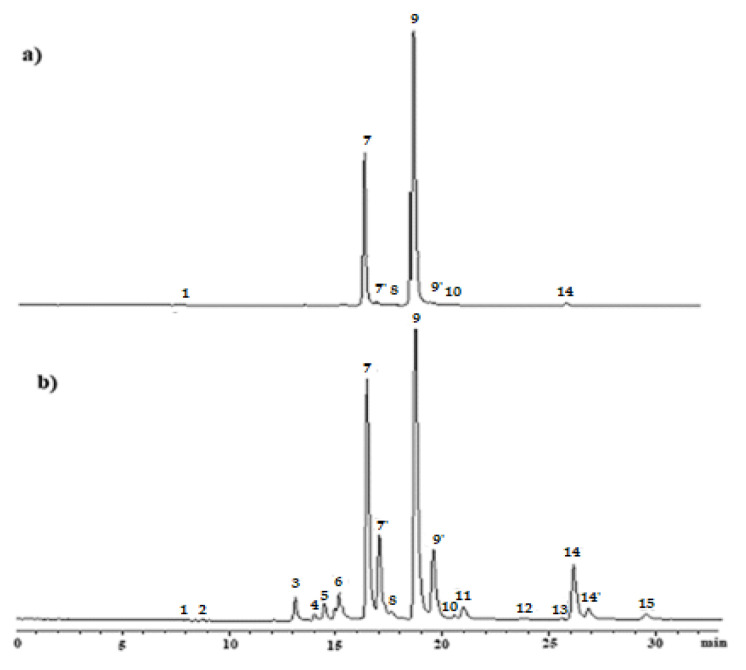
HPLC chromatograms at 640 nm of pigment extracts from: (**a**) fresh fruits; (**b**) fruits after alkaline treatment (sample A). Peaks: (1) pheophorbide *a*; (2) pyropheophorbide *a*; (3) Mg-15^2^-Me-phytyl-rhodin *g*_7_ ester; (4) 15^2^-Me-phytyl-rhodin *g*_7_ ester; (5) Mg-15^2^-Me-phytyl-chlorin *e*_6_ ester; (6) 15^2^-Me-phytyl-chlorin *e*_6_ ester; (7) chlorophyll *b*; (7′) chlorophyll *b*’; (8) 13^2^-OH-chlorophyll *b*; (9) chlorophyll *a*; (9′) chlorophyll *a*’; (10) 13^2^-OH-chlorophyll *a*; (11) Mg-15^2^-Me-phytyl-isochlorin *e*_4_ ester; (12) pheophytin *b*; (13) 15^2^-Me-phytyl-isochlorin *e*_4_ ester; (14) pheophytin *a*; (14′) pheophytin *a*’; (15) pyropheophytin *a*.

**Figure 2 foods-09-00701-f002:**
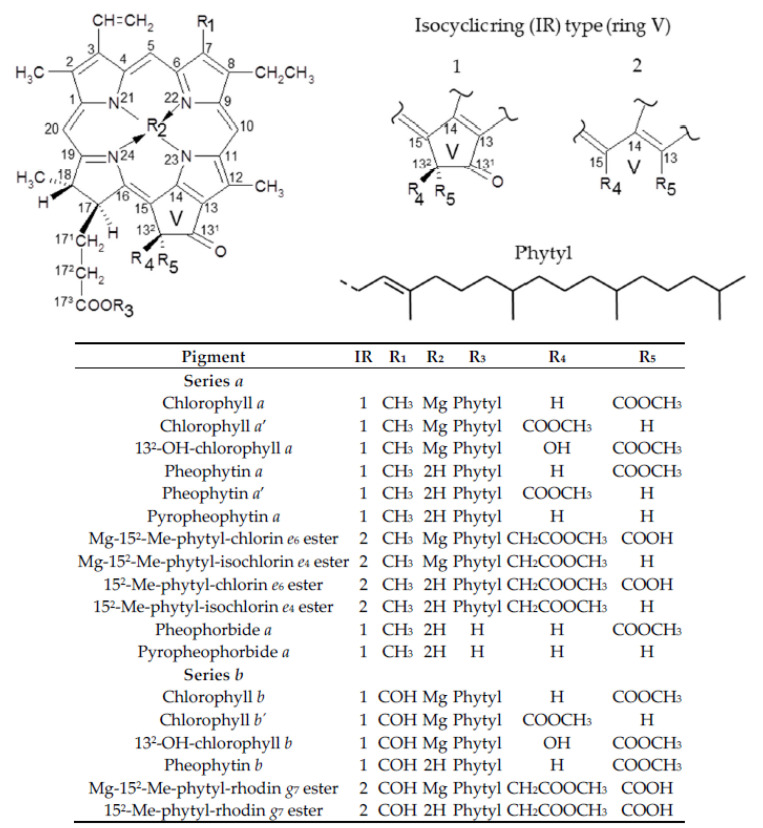
Structures of chlorophyll pigments present in green table olives.

**Figure 3 foods-09-00701-f003:**
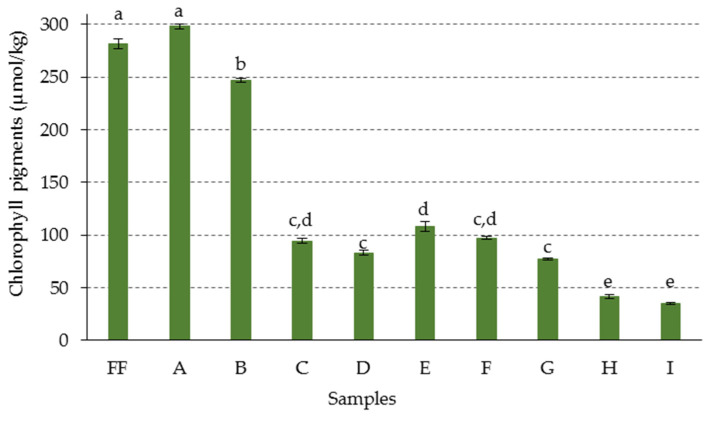
Total content (µmol/kg destoned fruit) of chlorophyll pigments in green olives. Abbreviations: FF fresh fruit; A–I: fruits with different alkaline treatments (see Table 1 for description of samples). Data represent mean values ± SD (n = 2). Different letters above the error bars indicate significant differences according to the Duncan’s multiple-range test (*p* < 0.05).

**Figure 4 foods-09-00701-f004:**
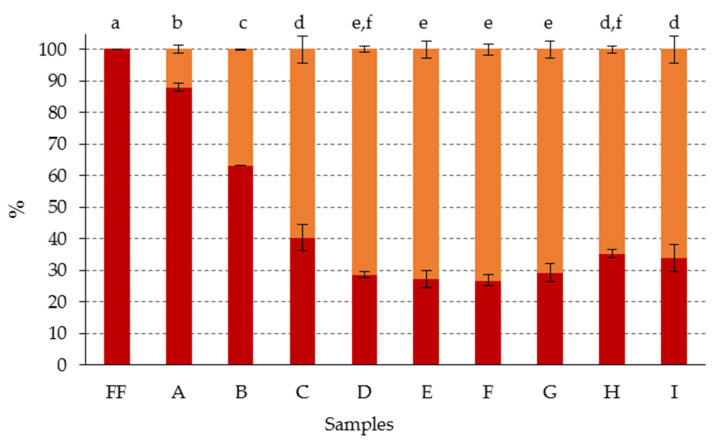
Percentage composition (with respect to total chlorophyll content) of chlorophyll pigments with: (●) closed isocyclic ring; (●) open isocyclic ring, in green olives. Abbreviations: FF, fresh fruit; A–I: fruits with different alkaline treatments (see Table 1 for description of samples). Data represent mean values ± SD (n = 2). Different letters above the error bars indicate significant differences according to the Duncan’s multiple-range test (*p* < 0.05).

**Figure 5 foods-09-00701-f005:**
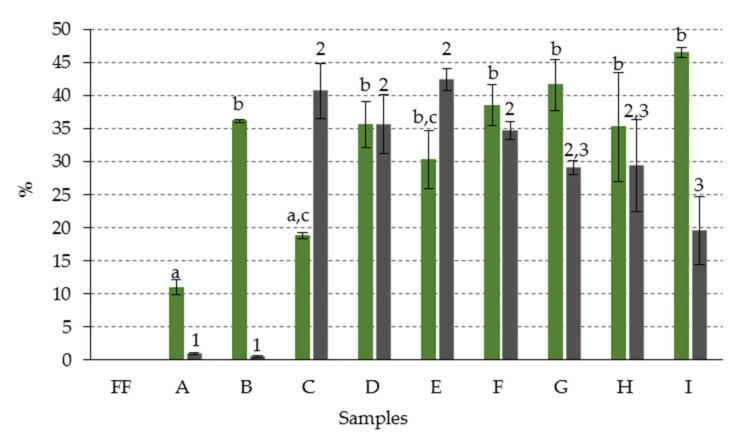
Percentage composition (with respect to total chlorophyll content) of chlorophyll derivatives with open isocyclic ring: (●) O-type; (●) Iso-type, in green olives. Abbreviations: FF fresh fruit; A–I: fruits with different alkaline treatments (see Table 1 for description of samples). Data represent mean values ± SD (n = 2). Different letters or numbers above the error bars indicate significant differences according to the Duncan’s multiple-range test (*p* < 0.05).

**Figure 6 foods-09-00701-f006:**
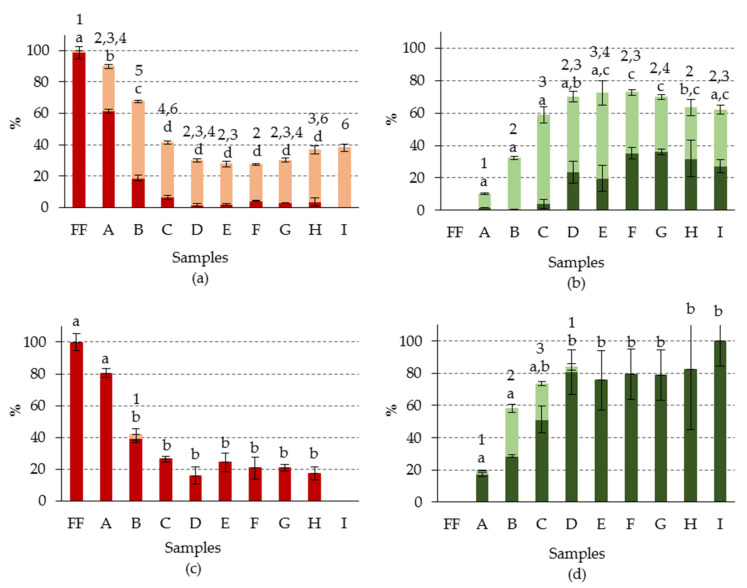
Percentage composition (with respect to total chlorophyll content) of chlorophyll pigments (●) with Mg and closed isocyclic ring; (●) Mg-free and with closed isocyclic ring; (●)with Mg and open isocyclic ring; (●) Mg-free and with open isocyclic ring, of (**a**,**b**): series *a*; (**c**,**d**): series *b*, in green olives. Abbreviations: FF, fresh fruit; A–I: fruits with different alkaline treatments (see Table 1 for description of samples). Data represent mean values ± SD (n = 2). Different letters or numbers above the error bars indicate significant differences (for the lower and upper data set, respectively) according to the Duncan’s multiple-range test (*p* < 0.05).

**Figure 7 foods-09-00701-f007:**
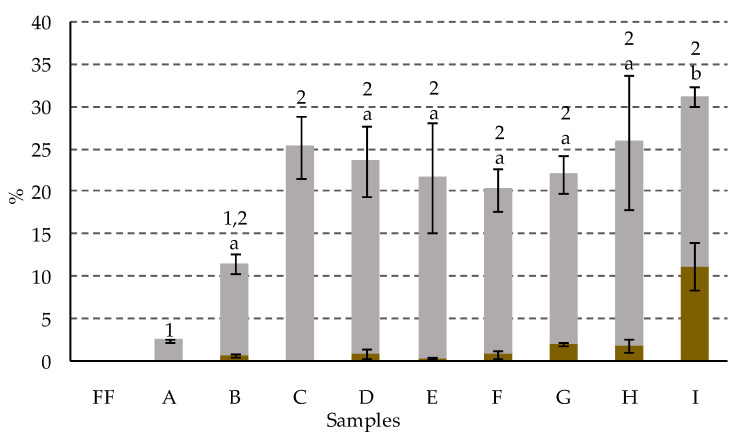
Percentage composition (with respect to total chlorophyll content) of: (●) pyropheophorbide *a*; (●) pyropheophytin *a*, in green olives. Abbreviations: FF, fresh fruit; A–I: fruits with different alkaline treatments (see Table 1 for description of samples). Data represent mean values ± SD (n = 2). Different letters or numbers above the error bars indicate significant differences (for the lower and upper data set, respectively) according to the Duncan’s multiple-range test (*p* < 0.05).

**Table 1 foods-09-00701-t001:** Sample codes and conditions (NaOH concentration and time) of the alkaline treatments applied to each sample.

Sample Code	A	B	C	D	E	F	G	H	I
**[NaOH] (%*w*/*v*)**	2	3	4	5	6	7	8	9	10
**Time (h)**	7	6	5	4	3	3	3	3	3

**Table 2 foods-09-00701-t002:** Chromatographic and spectroscopic characteristics in the HPLC eluent of chlorophyll pigments.

Pigments	Peak	t_r_ ^1^	k_c_’ ^2^	Spectroscopic Characteristics	
				Absorption Maxima (nm)	
	No			Soret	Q ^3^
**Series *a***					
Pheophorbide *a*	1	7.6	2.7	410	666
Pyropheophorbide *a*	2	8.9	3.3	410	666
Mg-15^2^-Me-phytyl-chlorin *e*_6_ ester	5	14.9	6.2	416	638
15^2^-Me-phytyl-chlorin *e*_6_ ester	6	15.1	6.3	400	662
Chlorophyll *a*	9	18.8	8.1	432	666
Chlorophyll *a*’	9’	19.6	8.5	432	666
13^2^-OH-chlorophyll *a*	10	20.2	8.8	434	664
Mg-15^2^-Me-phytyl-isochlorin *e*_4_ ester	11	21.0	9.2	416	638
15^2^-Me-phytyl-isochlorin *e*_4_ ester	13	26.1	11.7	400	662
Pheophytin *a*	14	26.3	11.8	410	666
Pheophytin *a*’	14’	26.8	12.0	410	666
Pyropheophytin *a*	15	29.6	13.4	410	666
**Series *b***					
Mg-15^2^-Me-phytyl-rhodin *g*_7_ ester	3	13.1	5.4	450	626
15^2^-Me-phytyl-rhodin *g*_7_ ester	4	14.0	5.8	426	650
Chlorophyll *b*	7	16.4	7.0	466	650
Chlorophyll *b*’	7’	16.9	7.2	466	650
13^2^-OH-chlorophyll *b*	8	17.6	7.5	466	646
Pheophytin *b*	12	24.0	10.7	436	654

^1^ t_r_: Retention time (min); ^2^ k_c_’: Retention factor = (tr − tm)/tm where tm is the retention time of an unretained component; ^3^ Q: maximum in the red region of the spectrum.

**Table 3 foods-09-00701-t003:** Chlorophyll pigment composition (μmol/kg destoned fruit) of green olives with different alkaline treatments ^1,2^. See Table 1 for description of samples.

Pigment	Fresh Fruit	Samples								
		A	B	C	D	E	F	G	H	I
Chlorophyll *a*	223.70 ± 13.17	115.60 ± 3.50	30.74 ± 5.43	5.10 ± 1.41	0.32 ± 0.08	0.56 ± 0.04	0.66 ± 0.02	0.34 ± 0.06	n. q.	n. q.
Chlorophyll *a’*	0.82 ± 0.07	31.95 ± 1.93	7.84 ± 1.52	0.70 ± 0.28	0.56 ± 0.34	1.34 ± 0.95	1.06 ± 0.08	1.78 ± 0.08	n. q.	n. q.
13^2^-OH-chlorophyll *a*	0.43 ± 0.06	n. q.	n. q.	n. q.	0.70 ± 0.63	n. q.	1.96 ± 0.28	n. q.	1.36 ± 0.97	n. q.
Mg-15^2^-Me-phytyl-chlorin *e*_6_ ester		1.46 ± 0.13	0.41 ± 0.29	n. q.	7.89 ± 0.43	7.64 ± 3.04	16.18 ± 3.52	14.22 ± 0.64	6.70 ± 0.46	6.78 ± 1.28
Mg-15^2^-Me-phytyl-isochlorin *e*_4_ ester		2.91 ± 0.39	1.19 ± 0.35	3.54 ± 2.22	9.87 ± 1.07	11.68 ±2.62	14.80 ± 2.50	10.88 ± 1.58	5.64 ± 6.06	1.82 ± 1.29
Pheophytin *a*	2.74 ± 0.25	49.43 ± 5.66	58.86 ± 1.10	4.36 ± 0.73	1.83 ± 0.22	1.84 ± 0.23	0.97 ± 0.22	1.83 ± 0.46	2.14 ± 0.44	1.00 ± 0.41
Pheophytin *a’*		11.52 ± 0.81	12.73 ± 1.18	2.69 ± 0.30	n. q.	n. q.	n. q.	n. q.	n. q.	n. q.
Pyropheophytin *a*		7.30 ± 0.34	26.44 ± 2.08	23.87 ± 0.73	18.82 ± 0.06	23.00 ± 1.94	18.74 ± 0.28	15.46 ± 1.04	9.94 ± 0.78	7.04 ± 0.40
15^2^-Me-phytyl-chlorin *e*_6_ ester		19.94 ± 1.44	64.72 ± 1.78	13.45 ± 1.49	15.26 ± 1.56	17.54 ± 2.28	14.32 ± 2.00	11.96 ± 1.28	5.66 ± 1.62	5.80 ± 0.94
15^2^-Me-phytyl-isochlorin *e*_4_ ester		n. q.	n. q.	34.94 ± 5.83	19.70 ± 2.70	34.16 ± 9.56	18.84 ± 0.66	11.58 ± 0.72	6.56 ± 1.92	5.10 ± 0.60
Pheophorbide *a*	0.20 ± 0.03	n. q.	0.49 ± 0.12	n. q.	n. q.	n. q.	n. q.	n. q.	n. q.	n. q.
Pyropheophorbide *a*		n. q.	1.83 ± 0.37	n. q.	0.76 ± 0.44	0.44 ± 0.08	0.86 ± 0.48	1.60 ± 0.00	0.78 ± 0.24	3.98 ± 0.98
Total series *a*	227.90 ± 6.44	240.10 ± 2.52	205.25 ± 2.06	88.65 ± 2.36	75.72 ± 2.60	98.20 ± 4.98	88.39 ± 1.62	69.65 ± 0.89	38.78 ± 2.37	31.52 ± 0.91
Chlorophyll *b*	52.99 ± 3.88	33.46 ± 1.92	11.14 ± 0.28	0.95 ± 0.05	n. q.	1.24 ± 0.60	n. q.	n. q.	n. q.	n. q.
Chlorophyll *b’*	0.97 ± 0.07	13.11 ± 0.39	5.20 ± 1.31	0.62 ± 0.11	n. q.	n. q.	0.20 ± 0.30	n. q.	n. q.	n. q.
13^2^-OH-chlorophyll *b*	n. q.	n. q.	n. q.	n. q.	1.22 ± 0.40	1.22 ± 0.42	1.60 ± 0.74	1.56 ± 0.02	0.48 ± 0.02	n. q.
Mg-15^2^-Me-phytyl-rhodin*g*_7_ ester		9.91 ± 0.64	11.85 ± 0.22	3.03 ± 0.48	6.12 ± 0.94	7.62 ± 1.68	6.86 ± 1.18	5.90 ± 1.02	2.26 ± 0.88	3.82 ± 0.42
Pheophytin *b*		n. q.	1.01 ± 1.43	n. q.	n. q.	n. q.	n. q.	n. q.	n. q.	n. q.
15^2^-Me-phytyl-rhodin *g*_7_ ester		1.44 ± 0.20	12.14 ± 1.01	1.31 ± 0.04	0.24 ± 0.16	n. q.	n. q.	n. q.	n. q.	n. q.
Total series *b*	53.96 ± 2.44	57.92 ± 1.04	41.34 ± 0.99	5.91 ± 0.25	7.58 ± 0.60	10.08 ± 1.06	8.66 ± 0.82	7.46 ± 0.72	2.74 ± 0.62	3.82 ± 0.42

^1^ Data represent mean values ± SD (n = 2); ^2^ n.q.: not quantified.

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
