# Peer review of "Influence of Alkaline Treatment on Structural Modifications of Chlorophyll Pigments in NaOH—Treated Table Olives Preserved without Fermentation"

_foods, 2020, doi:10.3390/foods9060701_

Round 1
Reviewer 1 Report
Alkaline treatment is a key stage in the production of green table olives and its main aim is rapid debittering of the fruit.However, chlorophyll pigments, which are located mainly in the skin of the fresh fruit could be affected. The authors studied the effect on chlorophyll pigments of nine different processes differentiated by NaOH concentration and/or treatment time.
Chlorophyll pigments analysis was performed by HPLC. Statistical Analysis was used for data processing using one–way analysis of variance (ANOVA)and post hoc comparisons were carried out according to Duncan's multiple range–test The authors found a direct relationship between the degradation of chlorophylls and the intenseness of the alkali procedure during the processing of green table olives.Quantified losses of 60% were observed at NaOH concentration of 4% or higher.
It is an interesting paper for scientists working in the domain as well as industrials. The manuscript is well written, protocol well developed and results treated carefully. Finally bibliography is up to date.
Author Response
We appreciate very much the reviewer' comments
Reviewer 2 Report
Dear Authors,Dear Editor,
The manuscript “Influence of alkaline treatment on structural modifications of chlorophyll pigments in NaOH–treated table olives preserved without fermentation” (Beatriz Gandul–Rojas, as corresponding author) for the Foods (Ref. foods-802441) analyses the effect of alkaline processes on chlorophylls (different NaOH concentration, treatment time), after one year of fruit preservation under refrigeration conditions. A direct relationship was found between the intensity of the alkali treatment and the degree of chlorophyll degradation Oxidation with opening of the isocyclic ring was the main structural change, followed by pheophytinization and degradation to colorless products.
In my opinion the manuscript is prepared in a professional manner, but it is not quite innovative in the subject. The authors continue the research focused on alkaline treatment in the production of green table olives. It is a valuable contribution to the already existing knowledge. The manuscript is very carefully prepared. The authors have extensive experience in the subject of research. The aim of the work is clearly established. Methods are adequate, the statistics well performed. It has a very exhaustive review of the literature and discussion of the results. The summary, introduction and discussion of the results are very, very detailed. Results are reliable and similar to those presented in the literature unfortunately by the same authors (their older research). Are there no results from other authors from other research centers? Conclusions are well-defined. In my opinion manuscript is acceptable for publication, but the use to discussion of 14 literature items (whose authors of this manuscript are co-authors (out of all 22 found throughout the discussion) is inappropriate.
Comments and suggestions
1)the drawback of the manuscript is using too many own literature items (in total 15); this should be corrected, replaced or supplemented with other publication; 2)the bibliography used in the manuscript not quite new (15-30 years ago); this should be corrected; 3)in many places in the manuscript, especially in the list of references there are numerous combined / stuck words;
Author Response
Response to comments of Reviewer 2 (Foods- 802441)
We appreciate very much the reviewer' comments and suggestions about the manuscript. We have tried to take into account all of them,however, there are some points that we would like to clarify in relation to them.
About the comments:
Point 1: Results are reliable and similar to those presented in the literature unfortunately by the same authors (their older research). Are there no results from other authors from other research centers?
Response 1: Most of our works have been dedicated to the study in depth of chlorophyll and carotenoid pigments in olive, table olive and olive oil, covering different aspects that may have an influence on each individual pigment, and in each product. Related to table olives, there are numerous works from other authors from other research centers, as well as from our own center, that determine the surface color of the olive by instrumental methods, but they do not study the pigments responsible for that color. We are pioneers in the study of the pigments of green table olives, and there are no studies from other researchers in the scientific literature, with the exception of the work of Aparicio-Ruiz et al., (2011), already included with number [44] in the original manuscript, and the ones of Negro et al., (2017) and Petigara et al., (2020) (this last so recently, that we have not had notice about it until now). These references are shown below and all of them are devoted to identification and quantification of Cu-chlorophylls complexes in commercial table olives.
- Aparicio–Ruiz, R.; Riedl, K.M.; Schwartz, S.J. Identification and quantification of metallo–chlorophyll complexes in bright green table olives by high–performance liquid chromatography–mass spectrometry quadrupole/time–of–flight. Agric. FoodChem.2011, 59, 11100–11108. https://doi.org/10.1021/jf201643s
- Negro, C.; De Bellis, L.; Sabella, E.; Nutricati, E.; Luvisi, A.; Miceli. Detection of not allowed food-coloring additives (copper chlorophyllin, copper sulphate) in green table olives sold on the Italian market. Adv. Hort. Sci., 2017 31(4) 225-233. https://doi.org/10.13128/ahs-20814
- Petigara Harp, B.; Scholl, P.F.; Gray, P.J.; Delmonte P. Quantitation of copper chlorophylls in green table olives by ultra- high-performance liquid chromatography with inductively coupled plasma isotope dilution mass spectrometry. Chromatogr. A 2020 461008. https://doi.org/10.1016/j.chroma.2020.461008
Related to the work of Aparicio-Ruiz et al., (2011),it is worthy to mention that this study resulted from a post-doc research of the first author, who was formed in our group under my supervision. In fact, the authors acknowledge me for reviewing the manuscript. As result of this work, a definitive structure of the allomerized chlorophyll compounds that our group had previously identified in table olives, were obtained by MS/MS data.
In the case of the work of Negro et al, (2017), its subject was far from our study and in the same way it is happening for the recently work of Petigara et al., (2020). Nevertheless, we have now introduced new sentences in the manuscript (lines 107-112 in revised manuscript) for including both references.
Point 2: Conclusions are well-defined. In my opinion manuscript is acceptable for publication, but the use to discussion of 14 literature items (whose authors of this manuscript are co-authors (out of all 22 found throughout the discussion) is inappropriate.
Response 2: We don´t fully understand this comment. We have checked the literature itemsused to discussion, and we have counted 5 from us (references [28,30,33,45,48]) of a total of 8 (the ones mentioned above, plus references [46,47,49]). In any case, as it has just been explained before, there are not found works in the bibliography from other authors, related with the subject discussed in the manuscript. Nevertheless, we have made an extra effort for including a new text with two additional citations (lines 268-277, and references [49] and [50] in revised manuscript).
Point 3: 1)the drawback of the manuscript is using too many own literature items (in total 15); this should be corrected, replaced or supplemented with other publication;
Response 3: The high number of self-citations in the manuscript was due to the explaining given in the first point; we are the only group dedicated to the study in depth of chlorophyll and carotenoid pigments in table olives. We can remove some of our references, if it is the preference of Reviewer 2, but without possibility to be corrected, replaced or supplemented with other publication, in those cases that they were included. The use of our own literatures is a reflection of reality in this subject that we cannot and must not hide.
Point 4: 2) the bibliography used in the manuscript not quite new (15-30 years ago); this should be corrected;
Response 4: In the original manuscript, from a total of 49 cited references, 32 are from the last 15 years, while 17 are older. Regarding to the comment of Reviewer 2, of course, we think that the literature must be up to date. However, we also think that if older studies complement ours, we must not forget them, especially, if there are no other researches related to their statements. Nevertheless, we have dispensed now with reference [13] (from 1995). On the other hand, an additional reason for the number of literatures from the 90s, was due to works related to chlorophyll and carotenoid pigments in table olives, which were necessary to refer to in the Introduction / background of the present study, for putting the reader in context.
Point 5: 3) in many places in the manuscript, especially in the list of references there are numerous combined / stuck words
Response 5: We regret about this. We have corrected them in the revised manuscript.
Reviewer 3 Report
In the present paper, the results of an analytical study on a specific variety of green table olives were showed. The authors describe the action that the alkaline treatment of olives has on the structural modifications of chlorophylls, bioactive components and pigments responsible for the highly appreciated green color of table olives.
The results allow to advance knowledge of the action of this treatment on Verdial variety (Olea europaea L).
The goals of the study can be considered achieved, even if the action that the alkaline treatment would become very complex if we considered all the variables on which it depends and worked on quantities greater than those used in the present study.
Excluding the treatment time as a variable, is a great advantage for the validity of the results. It should be better specified why and how.
Lines 144 – 146 specify how the reaching of the stone was established
Lines 146-147 “more concentrated alkaline solutions” specify ≥ 6%
Lines 266 – 267 I suggest to be more generic, even if a scientific study is mentioned
Lines 266 – 267 The Images may be superfluous; better to report only the table 2
Author Response
Response to comments of Reviewer 3 (Foods- 802441)
We appreciate very much the reviewer' comments and suggestions and according to them we have modifiedthe manuscript.
About the comments:
Point 1: Excluding the treatment time as a variable, is a great advantage for the validity of the results. It should be better specified why and how
Response 1: We have now included new sentences related to this comment (lines 330-335 in revised manuscript)
Point 2: Lines 144 – 146 specify how the reaching of the stone was established
Response 2: Information in relation to this comment has been now included (lines 150-152 in revised manuscript)
Point 3: Lines 146 – 147 “more concentrated alkaline solutions” specify ≥ 6%
Response 3: The sentence has been modified according to the reviewer´ suggestion (lines 152-153 in revised manuscript).
Point 4: Lines 266 – 267 I suggest to be more generic, even if a scientific study is mentioned
Response 4: We have included a generic paragraph in relation to chlorophyll pigment modifications in plant food processing (lines 268-272 in revised manuscript).
Point 5: Lines 266 – 267 The Images may be superfluous; better to report only the table 2
Response 5: At these lines there are not any images. We suppose that there is a mistake, and Reviewer is referring to Figure 1. But we are not totally sure about it and, therefore, we prefer not to remove Figure 1. In any case, we think that Figure 1 is of help for the discussion of the results.